# Detection of Human Papillomavirus Integration in Brain Metastases from Oropharyngeal Tumors by Targeted Sequencing

**DOI:** 10.3390/v13081536

**Published:** 2021-08-03

**Authors:** Brian McEllin, Brian C. Searle, Lisa DePledge, George Sun, Charles Cobbs, Mohsen Karimi

**Affiliations:** 1The Ben and Catherine Ivy Center for Advanced Brain Tumor Treatment, Swedish Neuroscience Institute, Seattle, WA 98122, USA; brian.mcellin@swedish.org (B.M.); lisadp@uw.edu (L.D.); george.sun@swedish.org (G.S.); charles.cobbs@swedish.org (C.C.); 2Department of Biomedical Informatics, The Ohio State University, Columbus, OH 43210, USA; searle.30@osu.edu

**Keywords:** brain metastasis, HPV, HNSCC, OPSCC, targeted sequencing, DNA target enrichment, data-independent acquisition, proteomics

## Abstract

Human papillomavirus (HPV) positive and negative head and neck squamous cell carcinoma (HNSCC) are known to have differential phenotypes, including the incidence and location of metastases. HPV positive (HPV+) HNSCC are more likely to metastasize to distant sites, such as the lung, brain, and skin. Among these locations, metastasis to the brain is a rare event, and little is known about specific risk factors for this phenotype. In this report, we describe two patients who developed brain metastases from HNSCC. Both patient tumors had p16^INK4a^ overexpression, suggesting these tumors were HPV+. This was confirmed after PCR, in situ hybridization, and mass spectrometry detected the presence of HPV type 16 (HPV16) DNA, RNA and protein. To further characterize the presence of HPV16, we used a target enrichment strategy on tumor DNA and RNA to isolate the viral sequences from the brain metastases. Analysis by targeted next generation sequencing revealed that both tumors had the HPV genome integrated into the host genome at known hotspots, 8q24.21 and 14q24.1. Applying a similar target enrichment strategy to a larger cohort of HPV+ HNSCC brain metastases could help to identify biomarkers that can predict metastasis and/or identify novel therapeutic options.

## 1. Introduction

Head and neck squamous cell carcinoma (HNSCC) affects more than 50,000 patients in the United States every year and accounts for over 10,000 deaths [1]. Over the past two decades, it has become clear that there are two subsets of HNSCC based on the presence or absence of human papillomavirus (HPV) [2]. The majority of these cases are associated with the high-risk HPV type 16 (HPV16) [2]. Patients with HPV positive (HPV+) HNSCC are predominantly male, tend to be non-smokers, and exhibit a better overall survival when compared to HPV negative (HPV−) HNSCC [3,4,5,6]. This survival has been attributed to better responses of the primary tumor to chemotherapy [3]. Although rates of metastasis are equal for both subsets [3,7], HPV− tumors tend to spread locally while HPV+ tumors tend to spread to distant sites such as lung, brain, and skin [8,9,10,11,12]. Among these sites, the brain is a relatively uncommon site for distant metastasis of HNSCC, accounting for only 7% of these metastases [11]. Only a limited number of studies have reported clinical data with HPV status for HNSCC brain metastases [8,10,12,13,14,15]. Although sample sizes are too small to make any definitive conclusions, together these may support a link between HPV and the development of brain metastases from the oropharyngeal subtype (OPSCC) of HNSCC. Among all the reported OPSCC cases with brain metastases and known HPV status, 27/33 of the tumors were HPV+.

While the presence of HPV in tumors can be used to stratify HNSCC patient populations, the evidence suggests the relationship between HPV and tumor phenotype may be more complex. For example, it has been hypothesized that transcriptional activity of HPV may be more important than HPV status alone in predicting the tumor phenotype. One study reported that transcriptionally inactive HPV+ HNSCC had similar gene expression patterns to HPV− tumors, and both showed worse progression free survival compared to HPV transcriptionally active tumors [16].

HPV integration into the host genome is an important step in HPV-induced tumorigenesis in cervical cancer [17] and may also promote HNSCC [18]. This is a relatively common event, occurring in 76% of HPV16+ cervical cancer and 72% of HPV+ HNSCC [19,20]. While a few initial studies did not find any relationship between HPV integration and survival in HNSCC [20,21], other reports have described specific circumstances when HPV integration had prognostic value. In one study, the authors demonstrated that HPV integration correlated with a worse prognosis only when hybrid viral-human sequences could be detected at the RNA level [22]. Another study compared tumors from treatment responsive vs. non-responsive HNSCC patients and found that these groups had different patterns of HPV integration—responsive tumors tended to have viral integration in intergenic regions, whereas non-responsive tumors had integration in cancer-associated genes [23]. These results underscore how a detailed understanding of the HPV molecular phenotype is essential to accurately predict outcomes.

To this point, limited research has examined the molecular characteristics of HPV in HNSCC brain metastases. Here, we provide a detailed genetic characterization of two cases of HPV16+ OPSCC brain metastases. These tumors were highly aggressive and featured metastases to lymph nodes and other sites. Multiple assays to detect HPV indicated that both patients had HPV16 positive tumors that expressed HPV transcripts. We were able to confirm integration of HPV16 into the human genome at two hotspots for HPV integration, 14q24.1 and 8q24.1. Similar to other studies with primary HNSCC [20], the HPV E1 gene appeared to be disrupted in both cases.

## 2. Materials and Methods

### 2.1. Tissue Collection

At the time of surgery, tumor tissue was removed and processed by Swedish Medical Center pathology services for paraffin embedding. Excess metastatic tissue (0.5–1 g) was transported to the laboratory on ice, flash frozen in liquid nitrogen, and stored at −80 °C. Paraffin sections were used for immunohistochemistry (IHC) and in situ hybridization (ISH) experiments. All patients were consented to participate in this study under our protocol approved by the WCG IRB (formerly Western IRB).

### 2.2. Immunohistochemistry for p16^INK4a^

Paraffin-embedded tissue sections (4 μm) were dewaxed in xylenes 3 times, passed through a series of graded alcohols, and washed in deionized water. For antigen retrieval, 1X CitraPlus AR Buffer (Biogenex) was pre-heated for 20 min in an Oster steamer before adding the slides for an additional 20 min. After cooling for 10 min, slides were washed 3 times in deionized water. Sequential blocking steps were performed—10 min in Bloxall (Vector Laboratories), 30 min in F_c_ Receptor Block (Biogenex) and 30 min in 1% normal horse serum (Vector Laboratories). Sections were incubated overnight at 4 °C in 1:200 rabbit IgG anti-CDKN2A/p16INK4a antibody (Abcam, ab108349). After washing 3 times in TBST, sections were incubated for 30 min in Vector ImmPRESS HRP Horse Anti-Rabbit IgG Polymer reagent. After washes, slides were developed for 3 min with Vector ImmPACT DAB. Sections were then counterstained with hematoxylin, dehydrated, cleared and mounted using VectaMount (Vector Laboratories). Images were captured using a Leica DM 2000 upright microscope.

### 2.3. RNAscope ISH for HPV16/18 E6/E7

Paraffin-embedded tissue sections were assayed using RNAscope probes from Advanced Cell Diagnostics (ACD) for HPV16/18 E6/E7 mRNA (#311121), PPIB (#313901, positive control), and dapB (#310043, negative control). No modifications were made to the described protocol for the Manual RNAscope Assay.

### 2.4. PCR Detection of HPV Nucleic Acids

Total genomic DNA and RNA were extracted from 10–20 µg of tissue using Qiagen DNeasy and RNeasy kits. PCR for HPV16 E6 and HPV18 L1 was performed on extracted DNA as described in Hashida et al. [24]. To detect E6 splice forms, a cDNA library was prepared from total tumor RNA using the High-Capacity cDNA Reverse Transcription Kit (Applied Biosystems), followed by PCR using the full length, E6*I, and E6*II specific primers described in Walline et al. [23].

### 2.5. SureSelect Target Enrichment for Viral Nucleic Acids

A target enrichment high-throughput sequencing based on the SureSelect technology (Agilent) was used to detect HPV encoded DNA and RNA in these tumors. SureSelect RNA baits were custom designed to detect 9 different viruses (HPV16, HPV18, HSV1, HSV2, HHV4, HHV5, HHV6A, HHV6B and HHV7). To achieve a high sensitivity, a high-resolution tilling design was applied with 5X coverage per region. Our custom designed SureSelect design contains 79,673 RNA biotinylated probes in total (Design ID: S3200262). Target enrichment reactions were performed using SureSelect XT kit. Enriched libraries were sequenced by Illumina sequencing platforms in a pool of 32 barcoded sample batches. FASTQ files were aligned to a reference sequence consisting of all targeted viruses (NC_001526.4) plus human genome GRCh38 by BWA for DNA [25] and STAR for RNA [26] sequences. Aligned reads were visualized by IGV Software.

### 2.6. Proteomics Sample Preparation

Protein samples were extracted with 5% SDS with 50 mM Triethylammonium bicarbonate, 2 M MgCl_2_, and HALT protease inhibitors (Thermo Scientific). Proteins were reduced with 20 mM dithiothreitol (DTT) for 10 min at 95 °C, alkylated with 40 mM iodoacetamide for 30 min at room temperature, and quenched with an additional 20 mM DTT. After alkylation, proteins were then acidified and loaded onto suspension traps (Protifi LLC), where they were digested with 1:25 sequencing grade trypsin (Promega) for 2 h at 47 °C. Resulting peptides were dried with vacuum centrifugation and brought to 1 μg/3 μL in 0.1% formic acid immediately prior to mass spectrometry acquisition.

### 2.7. Liquid Chromatography Mass Spectrometry

Tryptic peptides were separated with a Thermo Easy nLC 1000 and emitted into a Thermo Fusion Lumos using a 75 μm inner diameter fused silica capillary with an in-house pulled tip. The column was packed with 3 μm ReproSil-Pur C18 beads (Dr. Maisch) to 30 cm. A Kasil fritted trap column was created from 150 μm inner diameter capillary packed to 4.5 cm with the same C18 beads. Peptide separation was performed using 250 nL/min flow with solvent A as 0.1% formic acid in water and solvent B as 0.1% formic acid in 80% acetonitrile. For each injection, 3 μL (400–500 *m*/*z* to 800–900 *m*/*z* injections) or 5 μL (900–1000 m/z injections) was loaded and eluted using a two-step gradient: 35 min from 7 to 14% buffer B, followed by 55 min from 14 to 40% buffer B.

Following the approach described in Pino et al. [27], six gas-phase fractionated data independent acquisition (GPF-DIA) experiments were acquired of each sample (120,000 precursor resolution, 30,000 fragment resolution, AGC target of 4e5, max IIT of 60 ms, NCE of 33, +2H assumed charge state) using 4 *m*/*z* precursor isolation windows in a staggered window pattern with optimized window placements.

### 2.8. Mass Spectrometry Data Processing

Protein FASTA sequences for 9 viral species (HPV16, HPV18, HSV1, HSV2, HHV4, HHV5, HHV6A, HHV6B and HHV7) were downloaded from Uniprot on 29 May 2020, containing 1349 total entries. This FASTA database was digested in silico to create all possible +2H and +3H peptides between 7 and 30 amino acids with precursor m/z within 396.43 and 1002.70, assuming up to one missed tryptic cleavage. This peptide sequence and charge state list (37,119 total entries) was concatenated to a listing of 115,833 unique peptide sequence and charge state pairs that were detected in unpublished experiments of brain tumors and cell lines. Peptide fragmentation and retention time predictions for all peptides were made with the Prosit tool [28] and collected in a spectrum library using the approach presented in Searle et al. [29].

Data-independent acquisition (DIA) data were spectrum demultiplexed [30] with 10 ppm accuracy after peak picking in ProteoWizard version 3.0.18299 [31]. Library searches were performed using EncyclopeDIA version 0.9.5 [32], which was set to search with 10 ppm precursor, fragment and library tolerances, considering both B and Y ions and assuming trypsin digestion. Detected peptides were filtered to a 1% peptide-level false discovery rate. Fragmentation patterns from reported viral peptides were further required to have at least a 0.75 intensity correlation with the Prosit predictions.

## 3. Results

### 3.1. Case Presentation

#### 3.1.1. Case 1

A 69-year-old male patient was admitted to the hospital with a sore throat lasting 5 months. A laryngoscopy revealed an ulcerated tumor in the mid left of tongue extending toward the vallecular masses. Imaging and histological examination showed the mass was a stage T4aN2cM0 squamous cell carcinoma with p16^INK4a^ overexpression (Figure 1A,E,I). Surgery was performed to remove the primary tumor and involved lymph nodes, followed by concurrent radiation and chemotherapy. Around 16 months post-surgery, the patient began exhibiting progressive numbness and left sided hemi-paresis in the upper extremities. A magnetic resonance imaging (MRI) scan showed a 3.7 × 3.7 × 3.6 cm^3^ right parietal mass, which was subsequently surgically removed and confirmed to be a metastatic squamous cell carcinoma (Figure 1B,F). Five months later, the patient was treated concurrently with whole brain radiation and chemotherapy. Around 27 months post diagnosis, a positron-emission tomography (PET) scan revealed new multi-site metastases to the spinal cord, lung, and lymph nodes. Despite treatment with stereotactic radiosurgery and chemotherapy, the tumor continued to progress, and the patient passed away 30 months after initial diagnosis.

#### 3.1.2. Case 2

A 56-year-old man was admitted with a mass on the left side of his neck accompanied by pain in his head, neck, and shoulder. The mass had appeared one month before admission, and a PET scan showed a 3.4 × 2.3 × 2.6 cm^3^ mass within the left level 2 of the jugular digastric chain (Figure 1C). This was resected and determined to be a squamous cell carcinoma metastasis (Figure 1G). Subsequent imaging also revealed a mild asymmetry on the left side of tongue base. Histopathological examination showed a stage T1N3M0 high-grade squamous cell carcinoma with p16^INK4a^ overexpression. Lymph nodes were removed by surgery, but the original tongue tumor was inoperable at the time of presentation. The patient was treated with a combination of chemotherapy and radiation. At 8 months post diagnosis, a biopsy showed tumor spread to the paratracheal lymph node. This was followed by an additional course of radiation and chemotherapy. At 20 months post diagnosis, MRI screening revealed a <1 cm intra-axial mass in the left centrum with surrounding edema, which was then treated with Gamma Knife radiosurgery. At 33 months post diagnosis, the patient experienced episodes of confusion, and an MRI revealed that a 2.8 × 2.5 cm^2^ lesion had reappeared in the left centrum (Figure 1D). Surgery was performed to resect this mass, and histopathological examination showed this was a metastatic squamous cell carcinoma (Figure 1H). Within three months, there was evidence of tumor recurrence in the left frontal resection cavity. Over the next year, the patient underwent several rounds of Cyberknife, and after presenting with a metastasis to the optic nerve, the patient was placed in hospice. He passed away 57 months after initial diagnosis.

### 3.2. Molecular Analysis

We first analyzed formalin fixed, paraffin embedded (FFPE) tissue for evidence of HPV in all tumors (Case 1 primary tumor and brain metastasis; Case 2 neck lymph node metastasis and brain metastasis). Consistent with the presence of HPV, all four tumors displayed p16^INK4a^ overexpression by immunohistochemistry (Figure 1I–L). We next performed in situ hybridization on adjacent sections using the RNAscope HPV16/18 probe set. The results show HPV E6/E7 RNA was present in all samples in a pattern that directly matched the p16^INK4a^ overexpression (Figure 1M–P). In addition, DNA, RNA and protein were extracted from each patient’s metastatic brain tumor tissue. HPV DNA was assayed by real-time PCR using HPV16 E6 and HPV18 L1 Taqman assays, as reported previously [24]. Both cases were positive for HPV16 DNA and showed no evidence of HPV18 (Appendix A). HPV16 E6 is also known to have several splice variants, including E6*I and E6*II [33]. To see which variants were expressed, we performed PCR using a set of primers that can distinguish the full length E6, E6*I, and E6*II splice forms [23]. Both cases showed evidence of the E6*I and E6*II splice variants (Appendix A).

To further investigate the HPV molecular signature, we used a custom designed SureSelect kit (Agilent) to enrich for viral nucleic acids followed by next generation sequencing (NGS) on an Illumina Next Seq 500 system. Our kit contained probes covering the whole genomes of HPV16 and HPV18 as well as several herpesviruses. Resulting reads were aligned with BWA to a reference sequence consisting of all targeted viruses (with NC_001526.4 for HPV16) merged with human GRCh38/hg38. We detected HPV16 DNA in both cases, with no evidence of any other viruses above background reads. Case 1 showed coverage across most of the HPV16 genome, except for a 1500 bp deletion (NC_001526.4:643–2122) in the E1/E2 region (Figure 2A). This deletion appears to be linked to an integration event into the host genome at 14q24.1, as chimeric human-HPV DNA reads bordering this deletion were found spanning chr14:68,253,423—NC_001526.4:643 and NC_001526.4:2122—chr14:68,253,431 (Figure 2C). For Case 2, coverage was seen across the HPV16 genome without any major deletions, although some discontinuities were found (red and orange triangles, Figure 2B). As with Case 1, we observed evidence for HPV16 integration into the host genome bordering two of these discontinuities (red triangles, Figure 2B). Chimeric reads were found with junctions in the 8q24.21 locus at chr8:127649599(+)—NC_001526.4:1377(−) and NC_001526.4:3365(−)—chr8:127651202 (Figure 2D). Note that for both cases, the boundaries are approximate, as it was difficult to determine exact insertion boundaries. Both chromosomal locations have previously been reported as hotspots for integration in HPV associated cancers [34,35]. While no genes appear to be directly disrupted by these integration events, the integration at the 8q24.21 locus in Case 2 is upstream of the oncogene MYC. Similar instances of HPV16 and HPV18 insertions at 8q24 are associated with upregulation of MYC in cell lines from genital carcinomas [36].

We also sought to determine the extent of viral gene expression in Case 2 using the same SureSelect target enrichment strategy on cDNA libraries prepared from total tumor RNA. As expected from the DNA sequencing data, NGS data showed significant numbers of HPV16 reads and no HPV18 RNA reads. The highest expressed transcripts were the E1^E4, E6 and E7 genes, with comparatively little expression of E1 and E2 (Figure 3A). The sequencing also indicates that the dominant E6 transcript was the E6*I transcript (Figure 3C). We were also able to identify expression of the human-HPV fusion transcripts, confirming the insertion site into the host genome at 8q24.21 (Figure 3B).

To detect HPV encoded proteins, we analyzed protein lysates extracted from the metastatic tumor with tandem mass spectrometry. We searched the acquired data with EncyclopeDIA [32] using a spectrum library predicted with Prosit [28,29] from peptide sequences in the nine SureSelect viral genomes used above, concatenated to a human background. We observed confirmatory evidence for expression of five unique peptides from three HPV16 proteins (L1, E1 and E2) at a 1% false discovery rate (FDR), as demonstrated by extracted precursor and fragment ion chromatograms (Appendix A). These patterns additionally show high matching similarity to the Prosit fragmentation predictions, further increasing confidence in the match. E6 and E7 are short proteins (full length sequences are 19kDa and 11kDa, respectively) and may not produce any observable peptides. As expected, no peptides could be assigned to HPV18.

## 4. Discussion

Brain metastases from HNSCC carry a dire prognosis for patients, and the risk factors for their development are poorly understood. In this study, we present two cases of patients with HPV16+ OPSCC. Both patients developed brain metastases at 16 or 20 months after initial disease presentation, and neither patient had prior evidence of other distant metastases at this point. This time frame is within a typical range for HNSCC, as different studies have reported a range of median values between 17 and 36 months [10,13,14,37,38]. After the onset of a brain metastasis from HNSCC, the prognosis is poor. The largest two studies have reported a median survival between 2.5–3 months post brain metastasis [14,37], while one smaller study showed a median survival of 10.5 months [13]. However, both patients presented here survived longer (14 or 37 months) than the median survival from these studies.

We implemented a multifaceted strategy to characterize the molecular phenotype of HPV in these cases. Using FFPE tissue sections from the primary and metastatic tumors, we performed IHC for p16^INK4a^ and ISH for the E6/E7 genes from two common high-risk HPV strains, HPV16 and HPV18. Both assays supported the presence of HPV in both the initial and metastatic tumors, although we could not discriminate between strains with these assays. Subsequent assays were performed on DNA, RNA, and protein extracts from flash frozen metastatic brain tumor tissue. These indicated that only HPV16 was present in both tumors. We also determined that both metastatic tumors expressed HPV transcripts, including two E6 splice forms, E6*I and E6*II. This is important to identify, as active HPV transcription has been shown to have prognostic value in HPV+ OPSCC [16].

We next supplemented these results with target enrichment sequencing for HPV and human herpesviruses. This approach can be very powerful for molecular profiling of viruses in tumors, and it has already shown promise in identifying HPV sequences from human samples [39,40,41]. The ability to sequence HPV-enriched libraries from both tumor DNA and RNA provides several unique advantages over whole genome/transcriptome sequencing. Enriched libraries require much fewer reads per sample, allowing many tumors to be sequenced in a single run with a higher coverage depth. For these cases, targeted sequencing generated a robust data set that both replicated our initial data and provided additional insights. From targeted sequencing alone, we were able to identify the HPV strain, detect structural changes in the HPV genome, identify the gene expression patterns in tumors, and detect integration sites into the host genome. We were also able to rule out the presence of other oncogenic viruses such as EBV, which has been reported in some HNSCC [42].

We found that Case 1 had HPV integration at 14q24.1 and Case 2 had HPV integration at 8q24.21. Neither one of these directly disrupted human gene expression, and for Case 2, we were also able to show that hybrid human-viral transcripts from this integration were expressed. Understanding these details is important, as evidence suggests that HPV integration into the host genome can influence tumor phenotypes. When hybrid human-HPV transcripts were expressed, this correlated with a worse prognosis [22]. In addition, patients whose tumors responded to treatment exhibited integration into intergenic regions, whereas non-responsive tumors had integration into cancer-associated genes [23]. Furthermore, while structural similarities in different integration sites have been established [18], it is not clear how specific sites may impact tumor phenotypes. A recent meta-analysis of the literature identified 10 hotspots for HPV integration across > 1500 reported sites in cervical cancer and HNSCC [35]. While these were the most frequent sites, each individual site is relatively rare. Together, these ten most common integration sites comprised only 13% of the total integrations reported. Both tumors in this study had an integration event at one of these hotspots (14q24.1 and 8q24.21). Further study on a larger cohort of primary and metastatic OPSCC is necessary to determine if these sites have any correlation to metastatic phenotype.

In summary, we have presented detailed genetic characterizations of HPV16 in two OPSCC brain metastases. We have shown that each tumor was HPV16+ by multiple methods, including PCR, in situ hybridization, next generation sequencing and mass spectrometry. Both cases showed evidence of HPV integration into the host genome at known hotspots, although the significance of these sites is unknown. More comprehensive future studies using target enrichment strategies could help to identify specific commonalities in HPV integration among brain metastases. A more detailed understanding of this relationship could lead to new biomarkers to predict future metastasis and/or new treatment options.

## Figures and Tables

**Figure 1 viruses-13-01536-f001:**
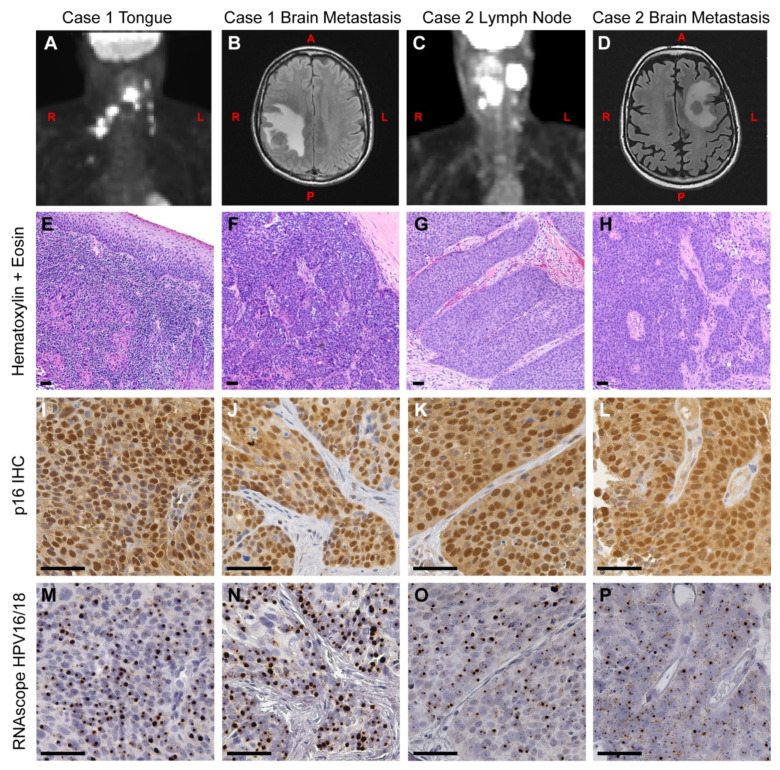
Clinical presentation and histology of primary and metastatic OPSCC. (**A**) A positron-emission tomography (PET) scan of Case 1 at the time of initial diagnosis, showing the primary tumor and spread to local lymph nodes. (**B**) A magnetic resonance imaging (MRI) scan was taken 16 months later, showing a right parietal mass in the brain. (**C**) A PET scan of Case 2 at the time of initial diagnosis, showing the neck mass and local spread to lymph nodes. (**D**) An MRI was taken 33 months later, showing a metastatic tumor in the left centrum. (**E**–**P**) Serial tissue sections from Case 1 (**E**,**F**,**I**,**J**,**M**,**N**) and Case 2 (**G**,**H**,**K**,**L**,**O**,**P**) were used for immunohistochemistry (IHC) and in situ hybridization analysis. (**E**–**H**) Primary and metastatic brain tumors were stained with hematoxylin and eosin. (**I**–**L**) Adjacent sections were assayed for p16^INK4a^ protein expression by IHC. All tumors displayed high levels of nuclear and cytoplasmic staining of p16^INK4a^, a hallmark of HPV+ tumors. (**M**–**P**) Tumors were assayed for HPV16 and HPV18 RNA using the RNAscope HPV16/18 probe set. The presence of HPV RNA is evident in both initial (**M**,**O**) and metastatic brain tumors (**N**,**P**). This pattern directly matches the pattern of p16^INK4a^ overexpression from the adjacent slides. L = left side, R = right side, A = anterior, P = posterior. Scale bar = 50 μm.

**Figure 2 viruses-13-01536-f002:**
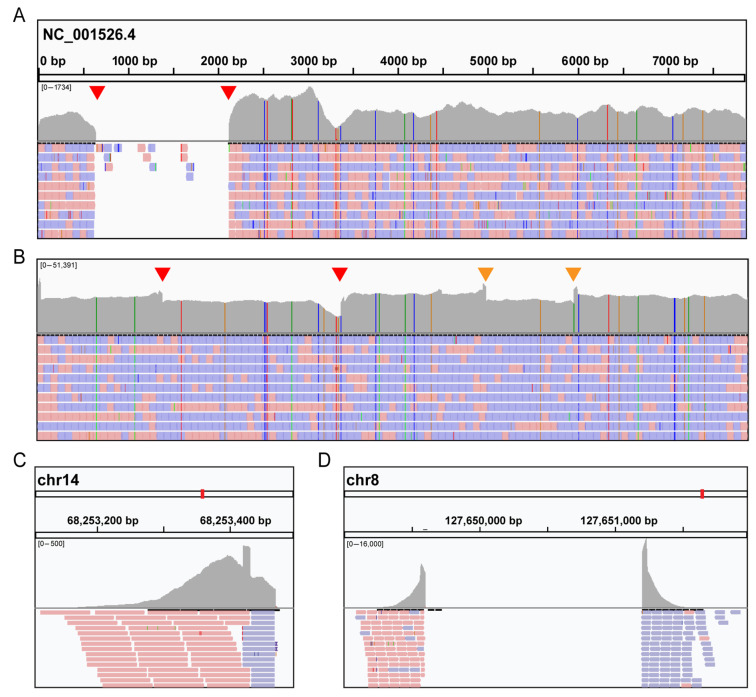
Characterization of human papillomavirus (HPV) integration in brain metastases by targeted sequencing. After sequencing the targeted libraries, reads were aligned to HPV16 reference genome (NC_001526.4) and visualized by IGV software. (**A**,**B**) Extensive coverage of the HPV genome was seen in both Case 1 (**A**) and Case 2 (**B**). Red arrows indicate discontinuities where chimeric reads between HPV and human genomes were detected. In addition, other discontinuities were detected in Case 2 (orange arrows). (**C**,**D**) In the lower panels, human DNA reads from chimeric HPV-human sequences are displayed, indicating an insertion site into the human genome at 14q24.1 (Case 1, (**C**)) and 8q24.21 (Case 2, (**D**)). Red (Forward) and blue (Reverse) colors correspond to specific strands.

**Figure 3 viruses-13-01536-f003:**
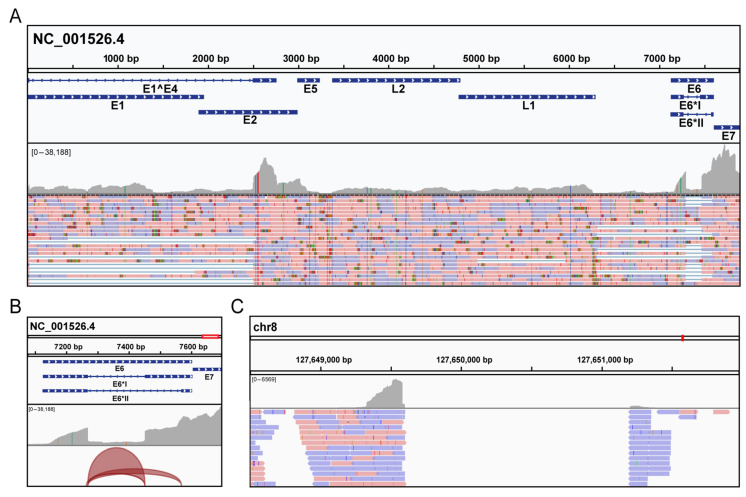
Detection of HPV mRNA in Case 2 by targeted sequencing. Total RNA was first extracted from the Case 2 brain metastasis, followed by separation of the mRNA fraction using oligo dT magnetic beads. This was followed by cDNA library preparation and enrichment of HPV encoded cDNAs using the SureSelect protocols. After Illumina sequencing, reads were aligned to HPV16 reference genome (NC_001526.4) and visualized by IGV software. (**A**) Data revealed a high level of E6 and E7 expression in the tumor. (**B**) The junction track showing alternative splice junctions in the E6 gene. The E6*I splice junction had more coverage (8368 reads) than E6*II (1561 reads). (**C**) Consistent with the DNA sequencing, human reads were detected for the integration site at 8q24.21, indicating the presence of a chimeric mRNA encoded from both HPV16 and the host genome.

## Data Availability

The genomics datasets used and/or analyzed during the current study are available from the corresponding author on reasonable request. Raw mass spectrometry datasets are available at MassIVE (MSV000086965, review password: “hpv16”) and through the ProteomeXchange (PXD024408).

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
