# Peer review of "Detection of Human Papillomavirus Integration in Brain Metastases from Oropharyngeal Tumors by Targeted Sequencing"

_viruses, 2021, doi:10.3390/v13081536_

Round 1

Reviewer 1 Report

In the manuscript by McEllin et al, the authors describe a two-case study of HPV assessment in brain metastases of HNSCC patients by multiple methods (IHC, RNA ISH, RT-PCR, NGS, mass spectrometry proteomic analysis).  Of interest is the fact that HPV was conclusively proven to be involved in the metastatic tissue.

The manuscript it limited with the number of samples (2 cases) which precludes generalizations, but is otherwise well written and as stated methodologically very robust. Some minor improvements could be made in the methods descriptions for clarity listed below

L74 what amount of excess frozen material was obtained for each case and how much was used for each subsequent method?

L100 why was E6 amplified for HPV16 and L1 for HPV18?

L113 “pool of 32 barcoded sample batches“. Considering there were only 2 samples what were the remaining 30 samples?

L148  unclear “detected in unpublished experiments of brain tumors and cell lines”. Was this done in your group or from other authors? Are there any published spectrometry data that you can validate your predicted database against?

L152 ”DIA data was demultiplexed“ suggests that multiple samples were run at once. However, since there was only 2 samples in total analyzed for this study was multiplexing necessary at all?

L231-  in the manuscript text the Case 1 is listed as having integration in 14q24 and Case 2 in 8q24 which is repeated in the Figure 2. However, at line 244 Case 1 is listed as being near c-Myc which is near 8q24

L256 why was only Case 2 selected for mRNA sequencing? If the reasoning was due to the integration near c-Myc, possibly figures should more clearly show the distance of the c-Myc and integration site?

L275-287. Comparing the Figure 3 (mRNA sequencing) and mass spectrometry data as presented in the manuscript is unusual. While not an expert in spectrometry, I would expect more fragments to be identified by spectrometry belonging to E7 and or E2 proteins than L1 protein? Some supplementary table showing other fragments identified would also be expected (even below the 1% treshold). As the results currently stand it appears that only a single peptide was seen by spectrometry which is highly unusual and raises the questions about method performance. Are there some “positive” controls, i.e. fragments that should be there that validate that the method worked in the first place and was analyzed successfully?  

Trivial/ Language

L76 „were consented for research“

L94-97 manufacturer of RNAscope probes should be stated for completeness

L104 typo  “Walline et al, [23].” Should be “et al.”

L219-220 „HPV16 E6 is also known to have several oncogenic splice variants“. It might be inappropriate to call splice variants oncogenic since their function is unknown and the whole length E6 protein is usually referred to as oncogenic

L243 „[34],[35].“ should be „[34, 35].“

Figure 1 subpanels are sometimes referred to with small letters (l198-206) and sometimes with capital letters (i.e. L215)

Supplemental material L14-15 – should be GAPHD primers instead of probes. Some reference or product number for the GAPDH primers would be helpful

Reviewer 2 Report

Dear Authors,

The manuscript is well structured and largely comprehensible, providing detailed genetic characterizations of two HPV related OPSCC brain metastases including evidence of HPV integration into the host genome at known hotspots. However, there are two main concerns to be addressed:

  1. HPV mRNA and protein was not analyzed in case 1 and the authors do not give a reason for this. The addition of these data would provide a more complete picture of the cases presented.
  2. The detection of the L1 protein by mass spectrometry (for case 2 only) is surprising because its expression is coupled to the differentiation process of HPV-infected skin and is unexpected for tumors. It remains an open question why the more important carcinogenic proteins E6 and E7 were not detected. Expression of L1 in brain tumor tissue should probably be verified by immunohistochemistry.

In addition, the following points should be considered to improve the manuscript:

Abstract and main text: “HPV16” is not defined. I suggest changing to “HPV type 16” to make it more clear or add some text introducing relevant HPV types.

There is usually no productive HPV replication and HPV-associated or -related cancers do not, in my opinion, meet the definition of "infection" (invasion and growth of germs in the body: NCI Dictionary of Cancer Terms) Therefore, this term should be avoided in reference to HPV-associated cancer.

Please ensure correct introduction of abbreviations at first mention and consistent use throughout the manuscript. E.g., there is a change from "HPV positive" to "HPV+" in the abstract; “DI water” in MM; also check for correct use of official gene and protein designations (e.g., p16).

I do not currently believe that patients with HPV-related HNSCC are younger, at least not in all populations. This should be put into perspective in the introduction and at least three of the four citations used are insufficient to support this: Ang et al. and Fakhry et al. both investigated selected study population of stage III or IV tumors and Gillison (2000) found that “The median age of patients with HPV-negative tumors was 64 years and was not significantly different from that of patients with HPV-positive tumors….”. There is evidence from epidemiologic data (e.g. PMID: 31358520) that the age incidence distribution peaks at 60 to 64 years of age (i.e. roughly comparable to HPV-negative OPSCC) and data from consecutively included OPSCC (e.g. PMID: 32624580) indicated that patients with HPV-related cancers are even older in some regions.

L.165: “(Figure 1A, E)” add Figure I (p16 staining). Check other references accordingly.

Supplementary Figure 2 is incorrectly labeled: the first lane should be called "M" and each marker band should be labeled according to its size.

Please check the results section for redundant information already mentioned in the MM, e.g., l. 276 and following.

MM/Discussion: Does HPV detection by tNGS detect and differentiate episomal HPV DNA? The absence of missing HPV sequences in case 2 could indicate the presence of episomal or concatameric HPV DNA. This should be investigated.

L.320: “We were also able to rule out any co-infection with other oncogenic viruses such as EBV” this should be clarified as the authors discuss sequencing of "HPV-enriched libraries" in this paragraph. Why were viruses other than HPV included in the sequencing analysis?

L.335-336: 2x together

Round 2

Reviewer 2 Report

No further comments.